# Chemical Reactivity Theory and Empirical Bioactivity Scores as Computational Peptidology Alternative Tools for the Study of Two Anticancer Peptides of Marine Origin

**DOI:** 10.3390/molecules24061115

**Published:** 2019-03-21

**Authors:** Juan Frau, Norma Flores-Holguín, Daniel Glossman-Mitnik

**Affiliations:** 1Departament de Química, Universitat de les Illes Balears, 07122 Palma de Mallorca, Spain; juan.frau@uib.es; 2Laboratorio Virtual NANOCOSMOS, Departamento de Medio Ambiente y Energía, Centro de Investigación en Materiales Avanzados, Miguel de Cervantes 120, Complejo Industrial Chihuahua, Chihuahua, Chih 31136, Mexico; norma.flores@cimav.edu.mx

**Keywords:** soblidotin, tasidotin, chemical reactivity theory, pKa, AGEs inhibition ability, bioactivity scores

## Abstract

This work presents an account of the reactivity behavior of the anticancer marine drugs, Soblidotin and Tasidotin, based on the calculation of the global and local descriptors resulting from Chemical Reactivity Theory (CRT), also known as Conceptual DFT, for their consideration as a useful complement to approximations based on Molecular Docking. The information on the global and local reactivity descriptors of the Soblidotin and Tasidotin molecules, obtained through our proposed methodology, may be used for the design of new pharmaceutical analogs by relying on the chemical interactions between these peptides and their protein-type biological receptors. It can be concluded that the CRT approximation to the global and local chemical reactivity, based on the descriptors, can provide interesting information for the consideration of both molecules as potential therapeutic drugs. This is complemented by a study on Advanced Glycation Endproduct (AGE) inhibition, by comparison with the usual molecular systems considered for the task, as a re-purposing study. Finally, the bioactivity scores for Soblidotin and Tasidotin are predicted through an empirical procedure, based on comparison with molecular structures with well-known pharmacological properties.

## 1. Introduction

There is considerable interest in finding new molecular systems that could be a basis for the design of novel therapeutic drugs. For the accomplishment of this objective, researchers have carried out searches for these potential therapeutic molecules in nature, based on experience acquired in recent decades. The sea is one of those places in which searches are carried out, as it is inhabited by an enormous variety of animal, plant, and microbial life, which generates diverse molecules and remains relatively unexplored.

Among the chemical species that can be found in marine life, peptides are one of the most important chemical groups. Peptides are molecules formed by the combination of natural amino acids and some of their variants, and represent ideal chemical species for the design of therapeutic drugs, since they possess interesting properties inherent to the molecular systems, ranging from small amino acids to large proteins, with which they are often compared.

Soblidotin, also known as Auristatin PE, has been investigated due its potential anti-tumor effects [1] and is a good example of a sea-derived peptide. According to the National Cancer Institute (NCI) Thesaurus, Tasidotin is a third-generation, synthetic, water-soluble, pentapeptide analog of the marine depsipeptide Dolastatin 15, with potential anti-mitotic and anti-neoplastic activities.

When one is working with Computational Chemistry and Molecular Modeling, there is not a universal methodology that could be applied to the entire spectra of known or unknown molecular systems. As such, our own devised methodology cannot be considered useful for all systems. The workaround that researchers in this field have found relies on studying, with a particular methodology, different but related families of molecules, in order to see if the proposed methodology can be applied with the same degree of success to the different group of molecules, and that the results obtained with a particular family of molecules are good not just by chance or some kind of serendipity. Indeed, the larger the number of studies with different groups of molecules, the greater the validity of the used methodologies.

Thus, the objective of this work is to perform a comparative study of the chemical reactivity of the Soblidotin and Tasidotin peptides, of marine origin, by resorting to Chemical Reactivity Theory (CRT) [2], in addition to an investigation of their potential AGEs inhibition abilities, following a protocol devised by our group [3,4,5,6,7,8,9,10,11,12]. Moreover, the bioactivity scores for Soblidotin and Tasidotin are predicted through an empirical procedure, based on comparison with molecular structures with well-known pharmacological properties.

## 2. Computational Methodology

Rather than resorting to the usual methodologies in Medicinal Chemistry, related to Molecular Docking, QSAR, and QSPR, we prefer to study the potential usefulness of the peptides by considering Computational Chemistry and Molecular Modelling, based on CRT and some semiempirical relationships relying on collected databases.

As with our recent studies [3,4,5,6,7,8,9,10], the computational tasks in this work have been done by considering the popular Gaussian 09 software [13]. Following the conclusions obtained from those studies, the MN12SX density functional [14] is chosen, because it can be considered to be well-behaved; according to our proposed, above-mentioned criteria. Accordingly, the calculation of the electronic properties used a model chemistry, based on the mentioned density functional in connection with the Def2TZVP basis set, while a smaller Def2SVP was considered for the prediction of the most stable structures [15,16]. In order to obtain accurate results, all calculations were performed using water as the solvent, simulated with the SMD model [17].

## 3. Results and Discussion

The molecular structures of the Soblidotin and Tasidotin peptides, which are depicted in Figure 1, were optimized in absence of solvent by resorting to the semiempirical DFTBA model (available in Gaussian 09), starting from the five most stable conformers selected from a pre-optimization accomplished by means of Molecular Mechanics techniques [18,19,20,21,22]. The resulting conformers were processed, as is customary within Computational Chemistry, to obtain the desired calculated properties, according to the techniques mentioned in the previous section.

The calculation of the maximum wavelength absorption of the Soblidotin and Tasidotin peptides was performed by conducting ground-state determinations with the same model chemistry. These results, together with the total, HOMO, and LUMO electronic energies, as well as the HOMO–LUMO gap, are shown in Table 1.

To get a glimpse of the electronic distribution around the atoms of both molecules, graphical sketches of the total electronic densities for Soblidotin and Tasidotin are presented in Figure 2.

### 3.1. Calculation of Global Reactivity Descriptors

The definitions for the chemical reactivity descriptors, coming from CRT, are provided in terms of the energies of the HOMO and LUMO, $\epsilon_{H}$ and $\epsilon_{L}$, respectively, as: The electronegativity $\chi \approx \frac{1}{2}\left(\epsilon_{L}+\epsilon_{H}\right)$ [23,24], the global hardness $\eta \approx (\epsilon_{L}-\epsilon_{H})$ [23,24], the electrophilicity $\omega \;=\;\approx \;{\left(\epsilon_{L}+\epsilon_{H}\right)}^{2}/4(\epsilon_{L}-\epsilon_{H})$ [25], the electrodonating power $\omega^{-}\approx {\left(3\epsilon_{H}+\epsilon_{L}\right)}^{2}/16\eta$ [26], the electroaccepting power $\omega^{+}\approx {\left(\epsilon_{H}+3\epsilon_{L}\right)}^{2}/16\eta$ [26], and the net electrophilicity $\Delta \omega^{\pm }\;=\;\omega^{+}\;+\;\omega^{-}$ [27]. The results of their calculations for the Soblidotin and Tasidotin peptides are presented in Table 2.

We can see, from Table 2, that, for both peptides, their electronating ability is more important that their electroaccepting ability, although there was not a significant difference between the values of the global reactivity descriptors of each of them.

### 3.2. Determination of the Local Reactivity Descriptors

If we now focus on the local reactivity descriptors coming from CRT, then the definitions will be: $f^{+}\left(r\right)=\rho_{N+1}\left(r\right)-\rho_{N}\left(r\right)$ for the Nucleophilic Fukui Function [23,24], $f^{-}\left(r\right)=\rho_{N}\left(r\right)-\rho_{N-1}\left(r\right)$ for the Electrophilic Fukui Function [23,24], and Δf(r) = ${\left(\partial \;f(r)/\partial \;N\right)}_{\upsilon (r)}$ for the Dual Descriptor [2,28,29,30,31,32], considering the electronic densities for a system with N+1, *N*, and N−1 electrons, respectively.

The Electrophilic Fukui functions $f^{-}\left(r\right)$ and Nucleophilic Fukui functions $f^{+}\left(r\right)$ for the anticancer marine peptides Soblidotin and Tasidotin are shown in Figure 3.

### 3.3. Calculation of the pKas of the Antimicrobial Peptides of Marine Origin and Quantification of the AGEs Inhibitor Ability

For the consideration of natural products to be useful for applications in Medicinal Chemistry, the knowledge of their pKas is very important. For large molecules, such as the peptides studied in this work, the experimental determination of their pKa is sometimes difficult. However, we have already shown that the pKa of a peptide can be predicted by applying a relationship with the global hardness η [11]. Thus, the values of the pKas of Soblidotin and Tasidotin, determined through that relationship, are presented in Table 3:

It is our belief that these results could be of interest for processes, involving drug design and development, using these peptides.

The Maillard reaction is, in fact, a collection of chained chemical reactions that start from the interaction between a reducing carbonyl and the free amino group of a peptide or protein. This chain of reactions leads to a final group of molecules, known as Advanced Glycation Endproducts (or AGEs), which are considered to be the main reasons for the development of some diseases, such as Diabetes, Alzheimers, and Parkinsons [33].

Many compounds have been devised as drugs with the goal of inhibiting the formation of AGEs, which include Pyridoxamine, Aminoguanidine, Carnosine, Metformin, Pioglitazone, and Tenilsetam [34,35]. Previous results from our group on the glycation process led us to conclude that the nucleophilicity N [36] may be a good descriptor for the AGE inhibition ability, and a nice definition for it could be based on the the inverse of the net electrophilicity $\Delta \omega^{\pm }$.

If we consider that this relationship is valid, then, based on the calculated global descriptors for Soblidotin and Tasidotin, it should be possible to predict their AGE inhibition ability, by comparison with the mentioned drugs:
ALT-946 > Aminoguanidine > Metformin > Carnosine > Tasidotin >
> Soblidotin > Tenilsetam > Pyridoxamine > > Pioglitazone


It can be seen that the studied peptides possess AGE inhibitor abilities, which are lower than that of other molecular systems already on the market [34,35].

### 3.4. Bioactivity Scores

When looking for the potential therapeutic capacities of new molecules, an interesting approach is an empirical procedure based on comparison with the properties of molecular structures with well-known pharmacological properties. These values are known as Bioactivity Scores, and can be easily estimated by resorting to the free online Molinspiration software, as shown in Table 4.

The results from Table 4 can be interpreted by considering the factors for evaluation of the Bioactivity Scores, which are shown in Table 5.

## 4. Conclusions

The outcomes of this work have resulted in the determination of important information about the chemical reactivity of two peptides of marine origin with potential therapeutic properties as anticancer drugs, Soblidotin and Tasidotin. This has been done by resorting to Chemical Reactivity Theory and some empirical relationships for the prediction of the pKa values, for their AGE inhibition abilities, and for the prediction of their Bioactivity Scores (as a measure of bioactivity).

The total electronic densities for both peptides have been displayed in the form of graphical sketches, as well as the HOMO and LUMO orbitals, as an approximation to the electrophilic and nucleophilic Fukui functions.

All these computed values could be of interest in the design and development of new pharmaceutical drugs, as well as for the practice of Medicinal Chemistry.

## Figures and Tables

**Figure 1 molecules-24-01115-f001:**
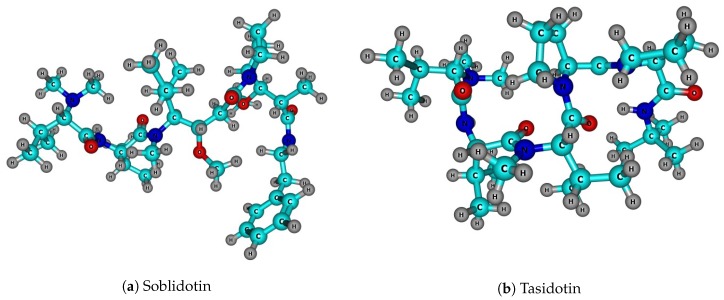
Graphical sketches of the molecular structures of the anticancer marine peptides: (**a**) Soblidotin and (**b**) Tasidotin.

**Figure 2 molecules-24-01115-f002:**
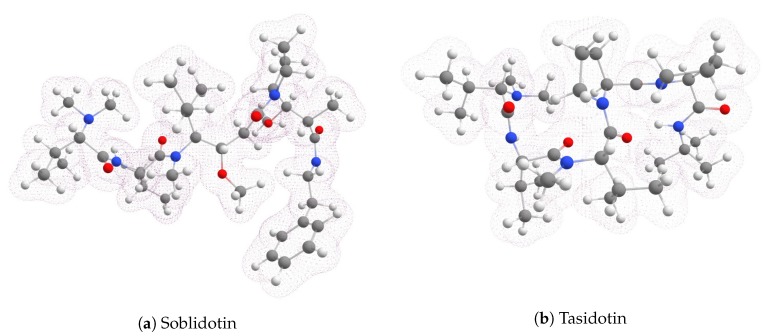
Graphical sketches of the total electronic densities of the anticancer marine peptides: (**a**) Soblidotin and (**b**) Tasidotin.

**Figure 3 molecules-24-01115-f003:**
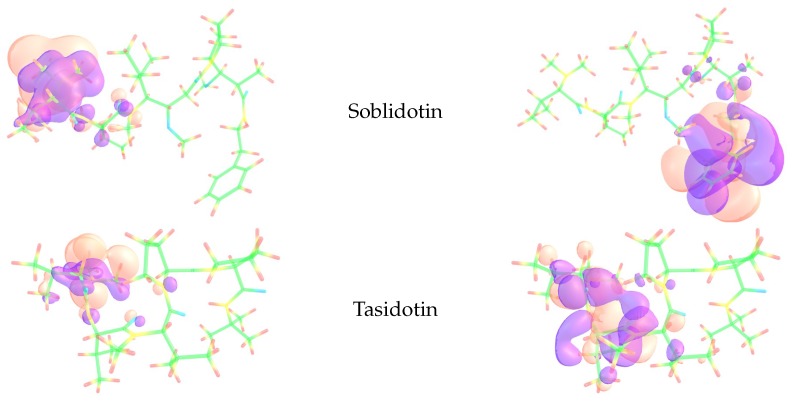
Graphical sketches of the HOMO and LUMO of the Soblidotin and Tasidotin marine peptides as approximations to the electrophilic (**left** column) and nucleophilic (**right** column) Fukui functions.

**Table 1 molecules-24-01115-t001:** Electronic energies of the neutral molecular systems (in au) of the anticancer marine peptides, Soblidotin and Tasidotin; the HOMO and LUMO orbital energies, as well as the HOMO–LUMO gap (in eV), and the maximum absorption wavelengths $\lambda_{max}$ (in nm).

Molecule	Total Electronic Energy	HOMO	LUMO	HOMO–LUMO Gap	$\lambda_{\max}$
Soblidotin	−2250.5737	−6.3060	−0.7353	5.5707	223
Tasidotin	−1958.1339	−6.2790	−0.6490	5.6300	220

**Table 2 molecules-24-01115-t002:** Global reactivity descriptors of the anticancer marine peptides Soblidotin and Tasidotin.

Molecule	Electronegativity	Global Hardness	Electrophilicity
Soblidotin	3.5206	5.5707	1.1125
Tasidotin	3.4640	5.6300	1.0657
Molecule	Electrodonating Power	Electroaccepting Power	Net Electrophilicity
Soblidotin	4.3334	0.8128	5.1463
tasidotin	4.2152	0.7512	4.9964

**Table 3 molecules-24-01115-t003:** pKa values for the anticancer marine peptides Soblidotin and Tasidotin.

Molecule	pKa
Soblidotin	11.70
Tasidotin	11.65

**Table 4 molecules-24-01115-t004:** Empirical Bioactivity Scores of the anticancer marine peptides, Soblidotin and Tasidotin.

Molecule	GPCR Ligand	Ion Channel Modulator	Kinase Inhibitor	Nuclear Receptor Ligand	Protease Inhibitor	Enzyme Inhibitor
Soblidotin	−0.30	−1.30	−1.07	−1.34	0.19	−0.83
Tasidotin	0.19	−0.43	−0.23	0.36	0.51	−0.13

**Table 5 molecules-24-01115-t005:** Factors for Evaluation of the Empirical Bioactivity Scores.

Active	Moderately Active	Inactive
>0	Between −5.0 and 0.0	<−5.0

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
