# Peer review of "Chemical Reactivity Theory and Empirical Bioactivity Scores as Computational Peptidology Alternative Tools for the Study of Two Anticancer Peptides of Marine Origin"

_molecules, 2019, doi:10.3390/molecules24061115_

Round 1
Reviewer 1 Report
The manuscript entitled “Computational Prediction of the Chemical Reactivity Properties of the Anticancer Marine Drugs Soblidotin and Tasidotin by Means of Conceptual DFT” by Juan Frau, et al, studied the chemical reactivity property of Soblidotin and Tasidotin and its contribution to their anticancer activity. Overall, the scientific content of the manuscript is poorly written and very difficult to understand.
1. The objective of the research work presented in this manuscript is not clear.
2. Authors mentioned/referred “our previous study” several times through the manuscript, in the methodology as well as results section, which is very annoying and it gives an impression of reading a review rather than a research article.
I would not recommend this manuscript for publication.
Author Response
1. The objective of the research work presented in this manuscript is not clear.
Response: We have taken into account all the comments posed by the Reviewer and modified our manuscript accordingly. Moreover, the manuscript has been entirely rewritten and the title has been changed from the original one presented at the moment of the former submission. As the Reviewer asked for extensive English editing, we have undergone this task by submitting our manuscript to the service offered by MDPI. A copy of the English-editing Certificate is attached to this submission. Lastly, we have submitted our revised manuscript to iThenticate in order to avoid overlap with other publications. The Similarity Report is also attached to this submission showing a Similarity Index of 0 %.
2. Authors mentioned/referred “our previous study” several times through the manuscript, in the methodology as well as results section, which is very annoying and it gives an impression of reading a review rather than a research article.
Response: Same response as for Comment Nº 1.
Reviewer 2 Report
The authors present a very specific drug design process, focused on two anticancers peptides Soblidotin and Tasidotin. The work is part of a big campaign to study natural products using their own algorithms and descriptors, and reveal possible bioactive sites and propose novel mode of actions. However i find several problems in the current article. First of all the only novelty is the two marine drugs presented. The rest of the article is already presented on citations 4-13. It would be very supporting if there were also biological data or experiments to validate the theoretical procedure. The authors selected the global minimum structure from Molecular Mechanics to proceed to Quantum Mechanic optimization. Since the peptides are very flexible, i would suggest to select the top 5 local minima structures and run the Quanthum Mechanic Calculations. The proposed mode of action is based on a online platform “Molinspiration” and there is no obvious connection with the theoretical calculations presented. Additionally they mention about the chemical reactivity of those natural products. The peptides studied cant’ be reagents for new drugs, since the peptide bonds are very easy to break in any kind of reaction. If the authors are mentioning their reactivity to new protein/targets they should explain it more clearly. Other corrections/suggestions A sketch is missing to present the compounds Lines 40-48 need rephrase. Very long sentence with complicated meaning. Lines 54-62 need rephrase, on how they present their own work. The word “us” could be replaced by “our group” Table 3, the pKa value of Tasidotin is not correct. A dot is missing. Line 214 “studied peptides studied” need correction Section 2 is named Computational Methodology, although inside the text they mention “Settings and Computational Methods” (line 198) I would propose this article to be re-written, become more clear in order to fully explain the title “Computational Prediction of the Chemical Reactivity Properties”Author Response
1. The authors present a very specific drug design process, focused on two anticancers peptides Soblidotin and Tasidotin. The work is part of a big campaign to study natural products using their own algorithms and descriptors, and reveal possible bioactive sites and propose novel mode of actions.
However i find several problems in the current article.
First of all the only novelty is the two marine drugs presented. The rest of the article is already presented on citations 4-13. It would be very supporting if there were also biological data or experiments to validate the theoretical procedure. The authors selected the global minimum structure from Molecular Mechanics to proceed to Quantum Mechanic optimization. Since the peptides are very flexible, i would suggest to select the top 5 local minima structures and run the Quanthum Mechanic Calculations.
Response:The manuscript has been corrected in order to explain the procedure for the optimization of the structures of the peptides. Indeed, we have considered the selection of the top five minima structures prior to run the Quantum Chemistry calculations, and this is now included in the corrected version of the manuscript.
2. The proposed mode of action is based on a online platform “Molinspiration” and there is no obvious connection with the theoretical calculations presented.
Response: The title of the manuscript has been changed to reflect the connection with the Molinspiration online platform.
3. Additionally they mention about the chemical reactivity of those natural products. The peptides studied cant’ be reagents for new drugs, since the peptide bonds are very easy to break in any kind of reaction. If the authors are mentioning their reactivity to new protein/targets they should explain it more clearly.
Response: This has been corrected in the revised version of the manuscript.
4. Other corrections/suggestions
A sketch is missing to present the compounds
Response: The graphical sketches of the molecular structures are already included.
5. Lines 40-48 need rephrase. Very long sentence with complicated meaning.
Response: We have taken into account all the comments posed by the Reviewer and modified our manuscript accordingly. Moreover, the manuscript has been entirely rewritten and the title has been changed from the original one presented at the moment of the former submission. As the Reviewer asked for extensive English editing, we have undergone this task by submitting our manuscript to the service offered by MDPI. A copy of the English-editing Certificate is attached to this submission. Lastly, we have submitted our revised manuscript to iThenticate in order to avoid overlap with other publications. The Similarity Report is also attached to this submission showing a Similarity Index of 0 %.
6. Lines 54-62 need rephrase, on how they present their own work. The word “us” could be replaced by “our group”
Response: Same answer as in the previous comment.
7. Table 3, the pKa value of Tasidotin is not correct. A dot is missing.
Response: Same answer as in the previous comment.
8. Line 214 “studied peptides studied” need correction
Response: Same answer as in the previous comment.
9. Section 2 is named Computational Methodology, although inside the text they mention “Settings and Computational Methods” (line 198)
Response: Same answer as in the previous comment.
10. I would propose this article to be re-written, become more clear in order to fully explain the title “Computational Prediction of the Chemical Reactivity Properties”
Response: We have taken into account all the comments posed by the Reviewer and modified our manuscript accordingly. Moreover, the manuscript has been entirely rewritten and the title has been changed from the original one presented at the moment of the former submission. As the Reviewer asked for extensive English editing, we have undergone this task by submitting our manuscript to the service offered by MDPI. A copy of the English-editing Certificate is attached to this submission. Lastly, we have submitted our revised manuscript to iThenticate in order to avoid overlap with other publications. The Similarity Report is also attached to this submission showing a Similarity Index of 0 %.
Reviewer 3 Report
In this manuscript, Authors investigated the molecular properties and structure of a group of anticancer marine drugs Soblidotin and Tasidotin using Conceptual Density Functional Theory (CDFT). I think that this theme of manuscript is suitable for Molecules. The manuscript deserves to publish in Molecules after a correction. I would like to suggest introducing changes before publishing in Molecules. I have a few comments and suggestions:
1. For what reason you applied MN12SX density functional? Did you check another density functional? Is it possible to compare theoretical and experimental results for this functional? The same questions also apply to the used basis set.
2. What about the distribution of electron density? According to me, an Electron Localization Function (ELF) analysis should also be made.
3. At the beginning of the manuscript you use a shortcut CDFT with reference to Conceptual Density Functional Theory. In the following part of manuscript, there are shortcuts Conceptual DFT. Please revisit the manuscript and make uniform the abbreviations. Please also pay attention to other shortcuts.
So, in my opinion, the manuscript is suitable in Molecules after a minor revision.
Author Response
1. For what reason you applied MN12SX density functional? Did you check another density functional? Is it possible to compare theoretical and experimental results for this functional? The same questions also apply to the used basis set.
Response: As it is mentioned in the manuscript, the MN12SX density functional has been chosen because as it has been verified in our previous works, it allows to fulfill our proposed criteria, that is to verify an approximate but very accurate Koopmans behavior. Indeed many other functionals have been tried and the MN12SX is the one which can perform accurately according to this criteria. Regarding the basis set, our previous published studies indicate that it is the most adequate to this end.
2. What about the distribution of electron density? According to me, an Electron Localization Function (ELF) analysis should also be made.
Response: As this work is based on Chemical Reactivity Theory (or Conceptual DFT) we believe that the inclusion of analysis from different methodologies may introduce more confusion. However, graphical sketches of the electron densities for both peptides have now been included in the revised version of the manuscript so that the interest reader can have a glimpse of the electron distribution around the atoms of the studied molecules.
3. At the beginning of the manuscript you use a shortcut CDFT with reference to Conceptual Density Functional Theory. In the following part of manuscript, there are shortcuts Conceptual DFT. Please revisit the manuscript and make uniform the abbreviations. Please also pay attention to other shortcuts.
Response: We have taken into account all the comments posed by the Reviewer and modified our manuscript accordingly. Moreover, the manuscript has been entirely rewritten and the title has been changed from the original one presented at the moment of the former submission. As the Reviewer asked for extensive English editing, we have undergone this task by submitting our manuscript to the service offered by MDPI. A copy of the English-editing Certificate is attached to this submission. Lastly, we have submitted our revised manuscript to iThenticate in order to avoid overlap with other publications. The Similarity Report is also attached to this submission showing a Similarity Index of 0 %.
Round 2
Reviewer 1 Report
The revised manuscript is much readable. However, there is no novelty in this study and the scientific significance is not high.
Reviewer 2 Report
The authors answered all of my questions and the manuscript has been modified and thus, it can be accepted.